# Lysophosphatidic Acid Promotes the Expansion of Cancer Stem Cells via TRPC3 Channels in Triple-Negative Breast Cancer

**DOI:** 10.3390/ijms23041967

**Published:** 2022-02-10

**Authors:** Naoya Hirata, Shigeru Yamada, Shota Yanagida, Atsushi Ono, Yukuto Yasuhiko, Motohiro Nishida, Yasunari Kanda

**Affiliations:** 1Division of Pharmacology, National Institute of Health Sciences, Kanagawa 210-9501, Japan; n-hirata@nihs.go.jp (N.H.); shyamada@nihs.go.jp (S.Y.); s-yanagida@nihs.go.jp (S.Y.); yasuhiko@nihs.go.jp (Y.Y.); 2Pharmacological Evaluation Institute of Japan (PEIJ), Ibaraki 305-0031, Japan; 3Division of Pharmaceutical Sciences, Graduated School of Medicine, Dentistry and Pharmaceutical Sciences, Okayama University, Okayama 700-8530, Japan; atsushiono3@okayama-u.ac.jp; 4Graduate School of Pharmaceutical Sciences, Kyushu University, Fukuoka 812-8582, Japan; nishida@phar.kyushu-u.ac.jp; 5Exploratory Research Center on Life and Living Systems, National Institutes of Natural Sciences, Aichi 444-8787, Japan; 6National Institute for Physiological Sciences, National Institutes of Natural Sciences, Aichi 444-8787, Japan

**Keywords:** lysophosphatidic acid, cancer stem cells, transient receptor potential cation channel subfamily C member 3, nuclear factor of activated T cells, triple-negative breast cancer

## Abstract

Triple-negative breast cancer (TNBC) is a highly aggressive cancer for which targeted therapeutic agents are limited. Growing evidence suggests that TNBC originates from breast cancer stem cells (BCSCs), and elucidation of the molecular mechanisms controlling BCSC proliferation will be crucial for new drug development. We have previously reported that the lysosphingolipid sphingosine-1-phosphate mediates the CSC phenotype, which can be identified as the ALDH-positive cell population in several types of human cancer cell lines. In this study, we have investigated additional lipid receptors upregulated in BCSCs. We found that lysophosphatidic acid (LPA) receptor 3 was highly expressed in ALDH-positive TNBC cells. The LPAR3 antagonist inhibited the increase in ALDH-positive cells after LPA treatment. Mechanistically, the LPA-induced increase in ALDH-positive cells was dependent on intracellular calcium ion (Ca^2+^), and the increase in Ca^2+^ was suppressed by a selective inhibitor of transient receptor potential cation channel subfamily C member 3 (TRPC3). Moreover, IL-8 production was involved in the LPA response via the activation of the Ca^2+^-dependent transcriptional factor nuclear factor of activated T cells. Taken together, our findings provide new insights into the lipid-mediated regulation of BCSCs via the LPA-TRPC3 signaling axis and suggest several potential therapeutic targets for TNBC.

## 1. Introduction

Breast cancer is the major cause of cancer death among women worldwide [1]. Triple-negative breast cancer (TNBC), which represents approximately 15% of all breast cancers, is characterized by the absence of estrogen receptor (ER), progesterone receptor (PR), and erb-b2 receptor tyrosine kinase 2 (ERBB2/HER2) expression [2]. In contrast to ER/PR- or HER2-positive subtypes, effective drugs for TNBC are limited to cytotoxic agents as molecular targets have not been elucidated. Therefore, new therapeutic strategies are required for TNBC.

Although the early detection and elucidation of the molecular mechanisms underlying breast cancer malignancy have progressed, cellular heterogeneity is considered to result in therapeutic resistance and promote cancer malignancy in many types of cancer, including breast cancer [3]. Growing evidence suggests that cancer stem cells (CSCs; also called tumor-initiating cells), a minor population of cancer cells with stem cell properties, are major drivers of cancer heterogeneity [4,5,6]. This minor population produces the bulk of cancer cells through continuous self-renewal and differentiation. CSCs have been isolated from diverse tumor samples, and established cell lines can be monitored using several methods, such as through monitoring cell surface marker expression or aldehyde dehydrogenase (ALDH) activity, side population analysis, and sphere-formation assays. ALDH assays are based on the fact that the expression of ALDH, a detoxifying enzyme responsible for the oxidation of intracellular aldehydes, is higher in stem cells than in differentiated cells [7,8]. ALDH1 expression is correlated with poor clinical prognosis in various cancers, including breast cancer [9]. Interestingly, TNBC tumors contain larger numbers of breast CSCs (BCSCs) than other types of breast cancer [10]. However, the molecular mechanisms that regulate the frequency and maintenance of BCSCs in TNBC remain poorly understood. 

Lysophosphatidic acid (LPA) is a lipid mediator that exerts multiple responses, such as proliferation, migration, and survival, via six G protein-coupled receptors (LPAR1–6) [11,12,13,14]. LPA is generated through the hydrolysis of lysophosphatidylcholine by autotaxin (ATX) [15] and degraded by several different enzymes, including lipid phosphate phosphatases (LPPs) [16]. The ATX-LPA axis has been considered to play a crucial role in cancer progression and resistance to chemotherapy [17]. 

Transient receptor potential cation (TRPC) channels have been implicated in diverse biological functions [18,19,20]. TRPC subfamily C member 3 (TRPC3) has been shown to regulate T cell receptor-mediated Ca^2+^ entry via nuclear factor of activated T cells (NFAT), a Ca^2+^-responsive transcription factor [21]. In addition, TRPC3 can promote cardiac hypertrophy through calcineurin and its downstream effector, NFAT [22]. 

Since autocrine and paracrine signaling play key roles in the maintenance and expansion of stem cells, we hypothesized that LPA plays a role in CSC regulation. We demonstrate here that LPA regulates BCSC expansion in TNBC. We also found that TRPC3, a calcium channel, is essential for LPA-induced proliferation of BCSCs via NFAT-mediated IL-8 secretion. Thus, these results implicate components of the LPA signaling pathway as therapeutic targets in TNBC.

## 2. Results

### 2.1. LPA Regulates the BCSC Population in MDA-MB-231 Cells via LPAR3

We have previously reported that the lysosphingolipid sphingosine-1-phosphate mediates the CSC phenotype, which can be identified as the ALDH-positive cell population in several types of human cancer cell lines [23]. To investigate whether additional lysosphingolipids regulate the frequency and maintenance of BCSCs in TNBC, we performed a quantitative polymerase chain reaction (qPCR) assay to detect lysosphingolipid receptors that were upregulated in ALDH-positive MDA-MB-231 cells. LPAR3 and LPAR6 expression was upregulated in ALDH-positive cells compared with ALDH-negative cells (Figure 1A). Since the expression level of LPAR3 in ALDH-positive cells was 13.2-fold higher than that of LPAR6 in ALDH-positive cells (Appendix A), we focused on LPAR3 in BCSCs. To investigate the involvement of LPA, we next examined the expression levels of LPA-related enzymes. As shown in Figure 1B, ATX, an enzyme involved in LPA synthesis, was also upregulated in ALDH-positive cells, while LPPs, which degrade LPA, were decreased. These results suggest a possible role for LPA metabolism in regulating CSC levels (Figure 1C).

We next examined whether LPA mediates CSC expansion. Stimulation with LPA increased the proportion of ALDH-positive cells in a dose-dependent manner, with a maximal response at 10 µM in MDA-MB-231 cells (Figure 2A). Similar effects of LPA were observed in another TNBC cell line HCC1806 (Figure 2A). Moreover, stimulation with LPA increased mammosphere formation in both TNBC cell lines: MDA-MB-231 cells and HCC1806 (Figure 2B). 

To examine the involvement of LPAR3, we used pharmacological antagonists against LPA receptors. The effects of LPA were blocked by an LPAR1/3 antagonist (Ki16425) but not by an LPAR2 antagonist (H2L5186303) in MDA-MB-231 cells (Figure 3A) and HCC1806 cells (Appendix A). To confirm the effect of LPAR3 on CSC expansion, we used a selective LPAR3 agonist (1-oleoyl-2-methyl-sn-glycero-3-phosphothionate; 2S-OMPT). As shown Figure 3B and Appendix A, 2S-OMPT caused an increase in ALDH-positive cells and mammosphere formation. Since Ki16425 inhibited both LPAR1 and LPAR3, we performed a knockdown experiment. LPAR3 siRNA inhibited an LPA-induced increase in ALDH-positive cells in MDA-MB-231 cells, whereas LPAR1 siRNA had little effect (Figure 3C). LPAR3 siRNA did not affect LPAR6 expression (Appendix A). These data suggest that LPAR3 mediates an LPA-induced increase in BCSCs. To confirm its involvement with the LPA receptor, we next examined G protein subtypes coupled to LPAR3. Pertussis toxin (PTX), which inactivates the G_i_ protein, abolished the LPA-induced increase in ALDH-positive cells in both MDA-MB-231 cells and HCC1806 cells (Figure 3D, Appendix A). The effects of PTX were confirmed by the overexpression of constitutively active mutants of G_i_ (Figure 3E). These results demonstrate that LPA increases BCSCs in TNBC via LPAR3/G_i_ signaling.

### 2.2. LPA Increases BCSCs via Calcium Signaling

Growing evidence suggests that there are many similarities between embryonic stem (ES) cells and CSCs; therefore, we next evaluated the Notch, Hedgehog, and Wnt signaling pathways as candidate pathways downstream of LPAR3. Stimulation with LPA did not induce expression of the Notch target gene hes family bHLH transcription factor 1 (HES1), the Hedgehog target gene GLI family zinc finger 1 (GLI1), or the Wnt target gene dickkopf WNT signaling pathway inhibitor 1 (DKK1) in MDA-MB-231 cells (Appendix A). In addition, self-renewal-related transcription factors Oct3/4, Nanog, Sox2, and c-Myc were not induced by LPA (Appendix A). We then examined other signaling pathways that are involved in ALDH-positive cells. Since LPARs are known to mobilize Ca^2+^ in ES cells [24], we examined calcium signaling. We found that LPA or 2SOMPT stimulation increased intracellular Ca^2+^, and the LPAR1/3 antagonist Ki16425 inhibited the LPA-induced increase in Ca^2+^ influx in MDA-MB-231 cells (Figure 4A). Calcium Ionophore A23187 was used to induce Ca^2+^ influx from outside as a positive control. Similar results were obtained with EGTA, which extracellularly inhibits Ca^2+^ influx from outside (Figure 4B). We next sought to identify involved calcium channels and focused on TRPC3, which plays a role in calcium homeostasis. As shown in Figure 4B, treatment with Pyr3, a selective inhibitor of TRPC3, inhibited the LPA-induced increase in Ca^2+^ influx. TRPC3 siRNA produced similar results. In addition, the LPA-induced increase in ALDH-positive cells was blocked by the membrane permeable Ca^2+^ chelator BAPTA-AM and Pyr3 in MDA-MB-231 cells (Figure 4C) and HCC1806 cells (Appendix A). Since TRPC3 was known to be activated by diacylglycerol, which was produced by phospholipase C, we examined the effect of phospholipase C inhibitor U73122 on an LPA-induced increase in ALDH-positive cells. As shown in Figure 4D, U73122 inhibited an LPA-induced increase in ALDH-positive cells in MDA-MB-231 cells. In addition, TRPC3 expression was upregulated in ALDH-positive cells compared with ALDH-negative cells (Figure 4E). TRPC3 siRNA inhibited the LPA-induced increase in ALDH-positive cells in MDA-MB-231 cells (Figure 4F).

To further examine the involvement of Ca^2+^ influx in regulating BCSC levels, we examined NFAT activation, which is downstream of the LPAR. LPA and 2S-OMPT stimulation increased the transcriptional activation of NFAT in MDA-MB-231 cells (Figure 5A). A23187 was used to induce NFAT activation as a positive control. The Ki16425 and NFAT inhibitor cyclosporin A (CysA) blocked the LPA-induced transcriptional activation of NFAT (Figure 5B) and the increase in ALDH-positive cells (Figure 5C). Similar effects of CysA were confirmed in HCC1806 cells (Appendix A). In addition, the overexpression of NFAT increased the number of ALDH-positive cells (Figure 5D). These results suggest that TRPC3 mediates the LPA-induced increase in ALDH-positive cells in TNBC via NFAT.

### 2.3. LPA Increases BCSCs via NFAT-Mediated IL-8 Production

We next investigated the transcriptional targets of NFAT by performing RNA sequencing (RNA-seq) analysis to investigate gene expression changes after LPA treatment of MDA-MB-231 cells. As shown in Figure 6A and Appendix A, we found that 428 transcripts were upregulated >2-fold and 420 transcripts were downregulated <0.5-fold among 15696 transcripts (>0.1 FPKM). In addition, we screened 207 protein-coding genes that were upregulated >2-fold. A gene ontology analysis revealed that upregulated protein-coding genes were categorized into specific signaling pathways, including those controlling inflammatory responses, drug response, and chemotaxis (Figure 6B). The possible NFAT-targeted genes included IL-8 because of the highest ranked candidate, which contains the target sequence of NFAT (Table 1) [25].

Real-time PCR confirmed that LPA induced IL-8 expression in MDA-MB-231 cells, and treatment with Ki16425, Pyr3, or CysA produced similar results (Figure 7A). LPA stimulation also increased IL8 secretion (Figure 7B), and this was inhibited by Pyr3 and CysA in MDA-MB-231 cells, suggesting that IL-8 is produced via an NFAT-mediated pathway. Furthermore, stimulation with IL-8 increased the number of ALDH-positive cells and mammosphere formation in MDA-MB-231 cells (Figure 7C and Appendix A). SB225002, a selective antagonist of the IL-8 receptor C-X-C motif chemokine receptor (CXCR)2, inhibited both IL-8- and LPA-induced increases in ALDH-positive cells (Figure 7D,E, Appendix A). In addition, IL-8, but not CXCR2, expression was upregulated in ALDH-positive cells compared with ALDH-negative cells (Figure 7F). Taken together, these results suggest that LPA induces and maintains the BCSC phenotype through the NFAT-mediated production of IL-8.

To assess the clinical significance of LPA signaling factors, we examined the gene expression levels in TNBC tissues and normal breast tissue. These RNA-seq data of breast cancer patients were obtained from the National Cancer Institute GDC data portal (https://portal.gdc.cancer.gov, accessed on 7 February 2022). Consistent with in vitro data of ALDH-positive cells in MDA-MB-231 (Figure 1A,C, Figure 4E, Figure 7A), the expression levels of LPAR3, TRPC3, and IL-8 were highly expressed and LPP1 and LPP3 were poorly expressed in TNBC tissues compared to normal breast tissues (Figure 8A,D,F–H). However, the expression levels of LPAR6, ATX, and LPP2 were different from ALDH-positive cells in MDA-MB-231 (Figure 8B,C,E). 

We then analyzed the effect of these genes on the prognosis of TNBC patients using the Kaplan–Meier plotter. As shown in Figure 9, the mRNA levels of LPAR3, TRPC3, and IL-8 were reversely correlated with the distant metastasis free-survival of TNBC patients. In contrast, LPAR6, ATX, and LPP1-3 had little effect on distant metastasis free-survival. These data suggest that TNBC tissues maintain the amount of LPA by inhibiting the degradation of LPA through the downregulation of LPP1 and LPP3. Thus, the LPAR3-TRPC3-IL-8 signaling axis might be clinically significant in TNBC.

## 3. Discussion

In this study, we used the ALDH assay to identify potential regulators of BCSC levels in TNBC and determined that LPA/LPAR3 signaling and subsequent TRPC3 activation resulted in increased BCSC numbers through the Ca^2+^-dependent transcriptional activation of IL-8 by NFAT (Figure 10). The findings presented here broaden our understanding of the role of lipid signaling in the development of breast cancer.

The results demonstrate that LPA has the ability to induce proliferation in TNBC. LPAR mRNAs are expressed in human ES cells [26], and LPA can increase the proportion of Oct3/4 and Nanog-positive cells in these cells [27], suggesting its importance in regulating stem cell populations. Since we previously reported that LPA had no effect on CSCs in the luminal breast cancer cell line MCF-7 [23], the effect of LPA on CSCs might be specific to TNBC cells. Thus, LPA might have self-renewal effects on CSCs and play a key role in stem cell regulation at some differentiation stage. Consistent with our data, a recent study showed that ovarian CSCs can be induced by LPA [28]. Therefore, LPA may play a general role in stem cell regulation. However, signaling events downstream of LPA are not well understood. In many cell types, LPA activates the mitogen-activated protein kinase (MAPK) and phosphatidylinositol 3-kinase/Akt pathways [29]. Whether these pathways are involved in the regulation of stem cells by LPA remains to be determined.

Our study indicates that LPAR3 is a novel target in BCSCs, which acted in concert with G proteins. We also found that G_i_ mediates the effects of LPA. LPA signaling is complex because different receptor subtypes are coupled to different G proteins. LPAR1-3s are coupled to G_i_ and G_q_, as well as G_12/13_ for LPAR1 and LPAR2, whereas LPAR4 and LPAR5 are coupled to G_q_ and G_12/13_. The effects of PTX suggest that the LPA-mediated BCSC expansion was selectively mediated by LPAR3 coupled to G_i_. Consistent with our data, LPAR3 expression is increased in human TNBC and is associated with tumor metastatic ability [30], suggesting its clinical relevance in breast cancer. LPAR3, therefore, represents a promising potential therapeutic target for TNBC treatment.

We also identified a novel role for TRPC3 in LPA/LPAR3-induced BCSC expansion using both selective inhibitors and siRNA. Although Orai1 has been shown to mediate LPA-induced Ca^2+^ influx and activation of NFAT2 [31], we found that siTRPC3 completely inhibited both an LPA-induced calcium entry and increase in ALDH-positive cells (Figure 4B,F), suggesting that TRPC3 mediated LPA-induced proliferation of BCSCs in TNBC. A role for TRPC channels in cancer growth has been reported in both prostate and ovarian cancer [32,33]. TRPC3 expression correlates with the differentiation grade of non-small cell lung cancer [34], and abnormal upregulation of TRPC3 and TRPC6 has been reported in breast cancer tissues [35]. In addition to TRPC3, previous reports have demonstrated low levels of TRPC4 and TRPC6 expression in immature stem cells [36], suggesting that the contributions of individual TRPCs may differ in different cell types or at different levels of differentiation. However, it is not well understood how TRPC family members regulate stem cell numbers and cancer progression.

Our study revealed that TRPC3 is oncogenic in MDA-MB-231 cells through the increased production of IL-8, an important molecule in cancer development. Clinical data analysis supports the prognostic role of TRPC3 and IL-8 in TNBC patients (Figure 9). The IL-8 receptors CXCR1/2 are also specifically overexpressed in BCSCs compared with bulk tumor cells, and these cells are resistant to Fas ligand-induced apoptosis [37]. In addition, a very recent report suggests that low IL-8 levels during chemotherapy are linked to prolonged survival [38]. However, the mechanisms by which IL-8 contributes to cancer progression are poorly understood. Our results suggest that high IL-8 levels induce the CSC phenotype in breast cancer. Therefore, IL-8 may mediate breast cancer malignancy by increasing the BCSC population. Blocking IL-8 signaling may provide a novel therapeutic approach for targeting CSCs.

Taken together, our results suggest that LPA-induced increases in BCSCs contribute to tumor growth in TNBC and delineate a signaling pathway by which the effects are mediated. In the future, it might be possible to establish specific treatments that reduce tumorigenesis by targeting the LPA/LPAR3 pathway and the TRPC3 channel. These novel therapeutic targets may someday result in the first clinically effective targeted therapies for TNBC.

## 4. Materials and Methods

### 4.1. Cell Culture

MDA-MB-231 cells (American Type Culture Collection) were cultured in Dulbecco’s modified Eagle’s medium (DMEM; Sigma-Aldrich, St. Louis, MO, USA) supplemented with 10% heat-inactivated fetal bovine serum (FBS; Biological Industries, Israel) and 100 U/mL penicillin and 100 µg/mL streptomycin (Gibco, Waltham, MA, USA). HCC1806 cells (American Type Culture Collection) were cultured in RPMI1640 medium (Gibco) supplemented with 10% heat-inactivated FBS, 100 U/mL penicillin, and 100 µg/mL streptomycin. For experiments, cells were seeded in phenol red-free medium containing 0.5% charcoal-stripped FBS. After 24 h, the medium was exchanged with phenol red-free medium containing 0.5% charcoal-stripped FBS with or without ligands.

### 4.2. Plasmid Constructs and Reporter Assays

Plasmids encoding NFAT were kindly provided from Shibasaki (Tokyo Metropolitan Institute of Medical Science, Tokyo, Japan). Plasmids encoding constitutively active mutants of G_i_ were kindly provided by J. Silvio Gutkind (National Institutes of Health, Bethesda, MD, USA). The luciferase activity was performed 24 h after transfection with NFAT-luc (Promega, #E848A, Madison, WI, USA) using the Dual Luciferase Reporter Assay System (Promega) according to the manufacturer’s instructions.

### 4.3. ALDH Assays

The ALDEFLUOR kit (Stem Cell Technologies, Vancouver, BC, Canada) was used to detect CSC populations with high ALDH enzyme activity, as previously described [39]. Cells (1 × 10^5^) were plated in 60 mm culture dishes. After serum starvation overnight, cells were stimulated once with LPA for 3 d. Cells were suspended at 1 × 10^6^ cells/mL in ALDH assay buffer containing the ALDH substrate BODIPY-aminoacetaldehyde (1 µM) and incubated for 30 min at 37 °C. ALDH-positive cells were measured by FACS Aria II cell sorter (BD Biosciences, San Jose, CA, USA). To establish gates of ALDH-positive and -negative cells, cells were treated with a specific ALDH inhibitor, diethylaminobenzaldehyde (15 µM), as a negative control. The same gates were applied to sort for ALDH-positive and -negative cells. After sorting of the ALDH-positive and ALDH-negative cells, these were immediately used in RNA isolation.

### 4.4. Mammosphere-Forming Assays

A sphere-forming assay was performed as previously described with slight modifications [38]. Briefly, cells were plated as single cells on ultra-low attachment 6-well plates (Corning, NY, USA) at a concentration of 2 × 10^4^ cells/well in 2 mL serum-free DMEM supplemented with N2 supplement (Gibco) and 20 ng/mL basic fibroblast growth factor (R&D Systems, Minneapolis, MN, USA) in the absence or presence of LPA. After 4 d, the number of mammospheres was counted on a microscope Olympus IX71 (Olympus, Tokyo, Japan).

### 4.5. Ca^2+^ Assay

Intracellular Ca^2+^ was measured using the Fluo-4 direct calcium assay kit (Thermo Fisher Scientific, Waltham, MA, USA) according to the manufacturer’s instructions. Briefly, MDA-MB-231 were plated at 1.5 × 10^4^ cells/well in 96 well culture plates the day before the assay. After serum deprivation for 2.5 h, the cells were incubated with Fluo-4 loading buffer for 30 min at 37 °C, then 30 min at room temperature. The cells were stimulated with LPA immediately before measurement. Each inhibitor was added for 30 min at room temperature before LPA stimulation. The fluorescence intensity was measured using a Fluoroskan Ascent FL microplate reader (Thermo Fisher Scientific) with excitation at 488 nm and emission at 515 nm.

### 4.6. Enzyme-Linked Immunosorbent Assay (ELISA)

MDA-MB-231 cells (5 × 10^5^) were plated in 60 mm dishes. After serum deprivation overnight, the cells were stimulated with LPA for 24 h. Each inhibitor was added for 30 min before LPA stimulation. Conditioned medium was harvested, and secretion of IL-8 was determined using the ELISA MAX Standard Set Human IL-8 (BioLegend, San Diego, CA, USA) according to the manufacturer’s instructions. The absorbance at 570 nm was measured using an iMark Microplate Reader (Bio-Rad, Hercules, CA, USA). The amount of IL-8 was calculated from a standard curve and normalized to the total protein content.

### 4.7. qPCR Analysis

Total RNA was isolated from MDA-MB-231 cells using TRIzol reagent (Thermo Fisher Scientific), and quantitative real-time reverse transcription (RT)-PCR was performed using the QuantiTect SYBR Green RT-PCR Kit (QIAGEN, Hilden, German) on an ABI PRISM 7900HT sequence detection system (Applied Biosystems, Foster City, CA, USA) as previously reported [40,41]. The relative change in the amount of transcript was normalized to the mRNA levels of glyceraldehyde-3-phosphate dehydrogenase (GAPDH). The primer sequences were shown in Appendix A.

### 4.8. Transient RNA Interference

TRPC3 and control siRNAs (Invitrogen, Waltham, MA, USA) were transfected into MDA-MB-231 cells using RNAiMAX (Invitrogen) according to the manufacturer’s recommendations.

### 4.9. RNA-Seq

Total RNA was isolated from MDA-MB-231 cells using the miRNeasy mini kit (Qiagen), and library construction and RNA-seq were performed at GENEWIZ (South Plainfield, NJ, USA). Sequencing data were analyzed using the Tophat/Cufflinks pipeline (Tophat v2.1.1; https://ccb.jhu.edu/software/tophat/index.shtml, accessed on 7 February 2022, Cufflinks v2.2.1; https://github.com/cole-trapnell-lab/cufflinks, accessed on 7 February 2022), and comparisons between control and LPA treatment cells were performed using Cuffdiff (v2.2.1; https://cole-trapnell-lab.github.io/cufflinks/cuffdiff/, accessed on 7 February 2022). Genes with low expression in both samples (average fragments per kilobase million <0.1) were filtered out.

### 4.10. Clinical Data

RNA-seq data of breast cancer patients were downloaded from the National Cancer Institute GDC data portal (https://portal.gdc.cancer.gov/, accessed on 7 February 2022). We selected 115 TNBC patient data from TCGA-BRCA dataset. Based on “ER status by IHC”, “PR status by IHC”, and “HER2 status by IHC”, we selected all negative status (ER, PR, HER2 negative) as TNBC patients. In addition, we obtained 38 available normal breast tissues tissue data. The effects of LPA signaling factors on distant metastasis free-survival of TNBC patients (ER-negative, PR-negative, and HER2-negative, *n* = 424) were analyzed using Kaplan–Meier Plotter (https://kmplot.com/analysis/index.php?p=background, accessed on 7 February 2022). The TNBC patient samples were divided into high expression and low expression groups by median.

### 4.11. Materials

LPA (18:1) (#857130P) was purchased from Avanti Polar Lipids, Inc (Alabaster, AL, USA) and dissolved in 50% ethanol. PTX (#168-22471) was purchased from Wako Pure Chemical (Osaka, Japan) and dissolved in water. Recombinant Human IL-8 (#200-08) was purchased from PEPROTECH (Rocky Hill, NJ, USA) and dissolved in water containing 0.1% BSA. 2S-OMPT (#10005707), Ki16425 (#10012659), and U-73122 (#70740) were purchased from Cayman Chemical (Ann Arbor, MI, USA). CysA (BML-A195) was purchased from Enzo Life Sciences (Farmingdale, NY, USA). H2L5186303 (#4878) was purchased from Tocris Bioscience (Bristol, UK). BAPTA-AM (#B035) solution (DMSO) was purchased from Dojindo Laboratories (Kumamoto, Japan). Pyr3 (#P0032), A23187 (#C7522), and SB225002 (#SML0716) were purchased from Sigma-Aldrich. These materials, other than LPA, PTX, and IL-8, were dissolved in DMSO. All other reagents were of analytical grade and obtained from commercial sources.

### 4.12. Statistical Analysis

Results are shown as the mean ± standard deviation (s.d.). Student’s *t*-tests were used to analyze the data, and *p* < 0.05 was considered significant. Statistical analyses were performed using Excel 2010 (Microsoft Corporation, Redmond, DC, USA) with the add-in software.

## Figures and Tables

**Figure 1 ijms-23-01967-f001:**
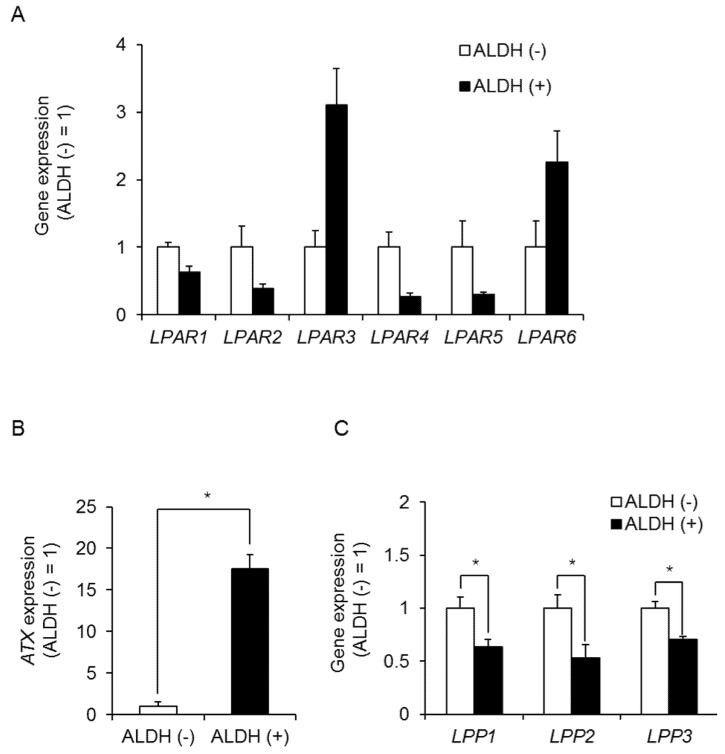
Role of the LPA receptor in ALDH-positive MDA-MB-231 cells. (**A**) LPA receptor expression in ALDH-positive and -negative MDA-MB-231 cells by qPCR. Data represent the mean ± s.d. (*n* = 3). (**B**) ATX expression in ALDH-positive and ALDH-negative cells in MDA-MB-231 cells by qPCR. Data represent the mean ± s.d. (*n* = 3). (**C**) LPP expression in ALDH-positive and ALDH-negative cells in MDA-MB-231 cells by qPCR. Data represent the mean ± s.d. (*n* = 3). * *p* < 0.05.

**Figure 2 ijms-23-01967-f002:**
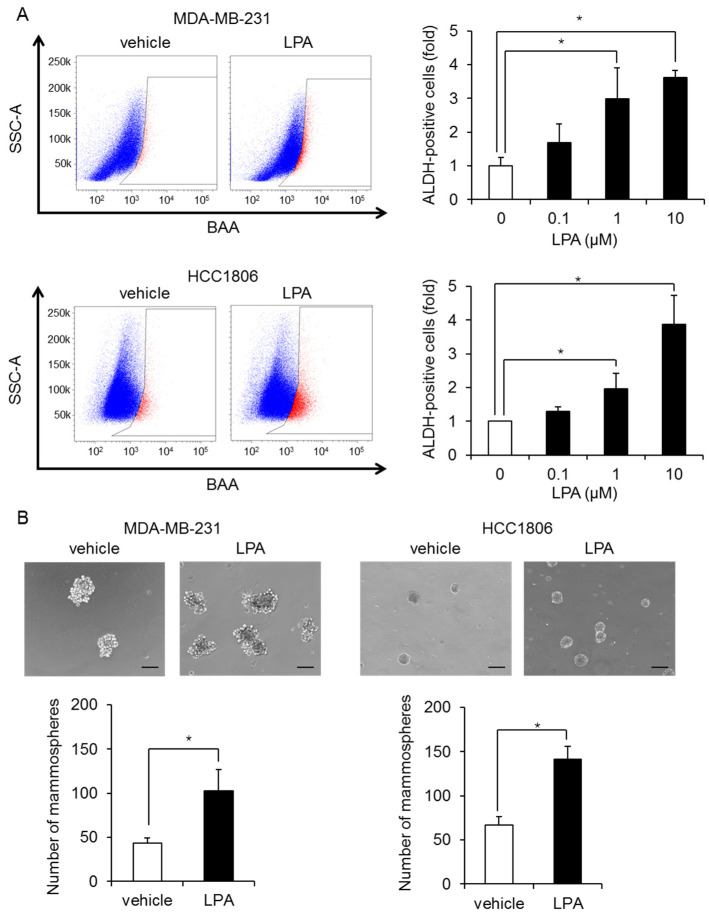
LPA increases the number of BCSCs in MDA-MB-231 cells and HCC1806 cells. (**A**) Representative flow data from MDA-MB-231 cells and HCC1806 cells treated for 3 days with or without 10 µM LPA (left). Dose-dependent effect of LPA on ALDH-positive cells (right). Data represent the mean ± s.d. (*n* = 3). (**B**) Effects of 10 µM LPA on the mammosphere-forming efficiency of MDA-MB-231 cells and HCC1806 cells. The number of mammospheres was counted using a microscope. The scale bar indicates 100 µm. Data represent the mean ± s.d. (*n* = 3). * *p* < 0.05.

**Figure 3 ijms-23-01967-f003:**
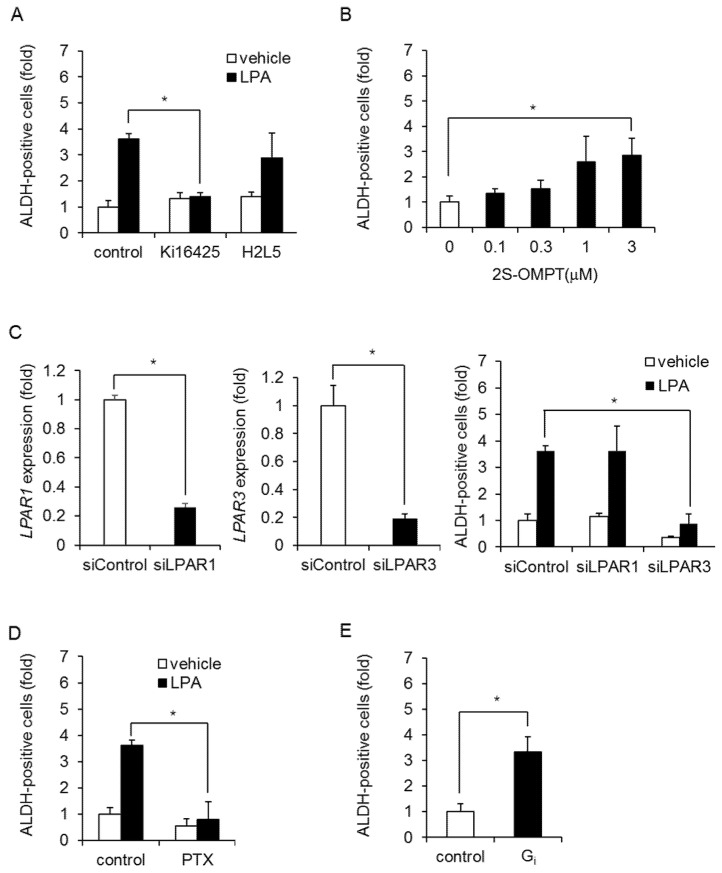
LPA increases ALDH-positive cells via LPAR3/G_i_ signaling in MDA-MB-231 cells. (**A**) Effects of LPAR1/3 and LPAR2 antagonists (Ki16425 and H2L5186303, respectively; both 10 µM) on the number of ALDH-positive cells after LPA treatment. Data represent the mean ± s.d. (*n* = 3). (**B**) Effect of the LPAR3 agonist 2S-OMPT on the number of ALDH-positive cells. Data represent the mean ± s.d. (*n* = 3). (**C**) Depletion of LPAR1 or LPAR3 with siRNA. Effect of LPAR1 or LPAR3 siRNA on the LPA-induced increase in ALDH-positive cells. Data represent the mean ± s.d. (*n* = 3). (**D**) Effect of PTX (100 ng/mL) on the number of ALDH-positive cells after LPA treatment. Data represent the mean ± s.d. (*n* = 3). (**E**) Effects of constitutively active mutants of G_i_ on the number of ALDH-positive cells. Data represent the mean ± s.d. (*n* = 3). * *p* < 0.05.

**Figure 4 ijms-23-01967-f004:**
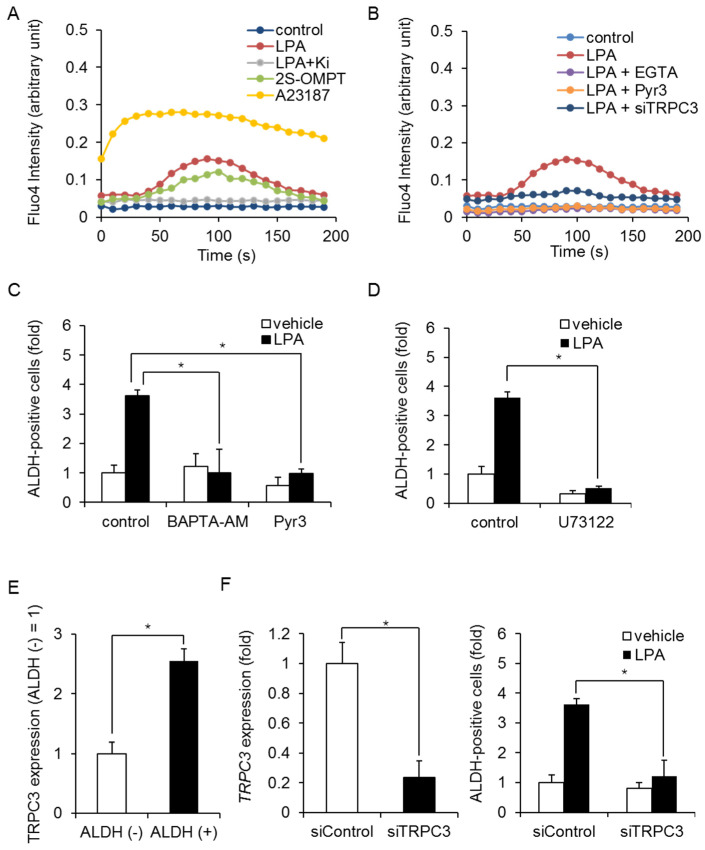
LPA-induced Ca^2+^ signaling in MDA-MB-231 cells. (**A**) After stimulation with LPA (10 µM) or 2S-OMPT (3 µM), intracellular Ca^2+^ in bulk cells was measured using Fluo-4. Effect of Ki16425 (Ki; 10 µM) on the LPA-induced increase in Ca^2+^ influx. A23187 (10 µM) was used as positive control. (**B**) Effects of EGTA (1 mM), Pyr3 (1 µM), and TRPC3 siRNA on the LPA-induced increase in Ca^2+^ influx. (**C**) Effects of BAPTA-AM (1 µM) and Pyr3 (1 µM) on the LPA-induced increase in ALDH-positive cells. Data represent the mean ± s.d. (*n* = 3). (**D**) Effect of U73122 (3 µM) on the LPA-induced increase in ALDH-positive cells. Data represent the mean ± s.d. (*n* = 3). (**E**) TRPC3 expression in ALDH-positive and ALDH-negative cells by qPCR. Data represent the mean ± s.d. (*n* = 3). (**F**) Depletion of TRPC3 with siRNA (left). Effect of TRPC3 siRNA on the LPA-induced increase in ALDH-positive cells (right). Data represent the mean ± s.d. (*n* = 3). * *p* < 0.05.

**Figure 5 ijms-23-01967-f005:**
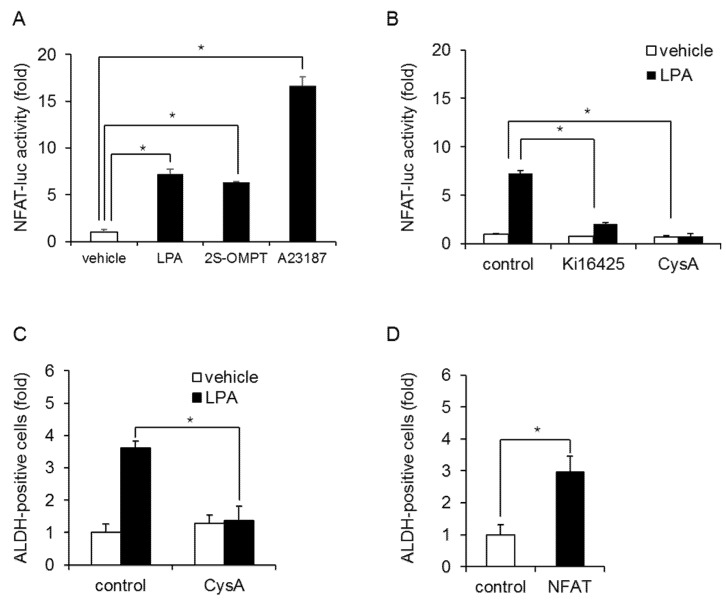
LPA-induced NFAT activation in MDA-MB-231 cells. (**A**) MDA-MB-231 cells transfected with a reporter plasmid encoding NFAT-luc were cultured with or without LPA (10 µM) or 2S-OMPT (3 µM) for 24 h and then analyzed by luciferase assays. A23187 (10 µM, 24 h) was used as positive control. Data represent the mean ± s.d. (*n* = 3). (**B**) Effect of Ki16425 (10 µM) or CysA (10 µM) on LPA-induced NFAT activation. Data represent the mean ± s.d. (*n* = 3). (**C**) Effect of CysA (10 µM) on LPA-induced increases in ALDH-positive cells. Data represent the mean ± s.d. (*n* = 3). (**D**) Effects of overexpression of NFAT on the number of ALDH-positive cells. Data represent the mean ± s.d. (*n* = 3). * *p* < 0.05.

**Figure 6 ijms-23-01967-f006:**
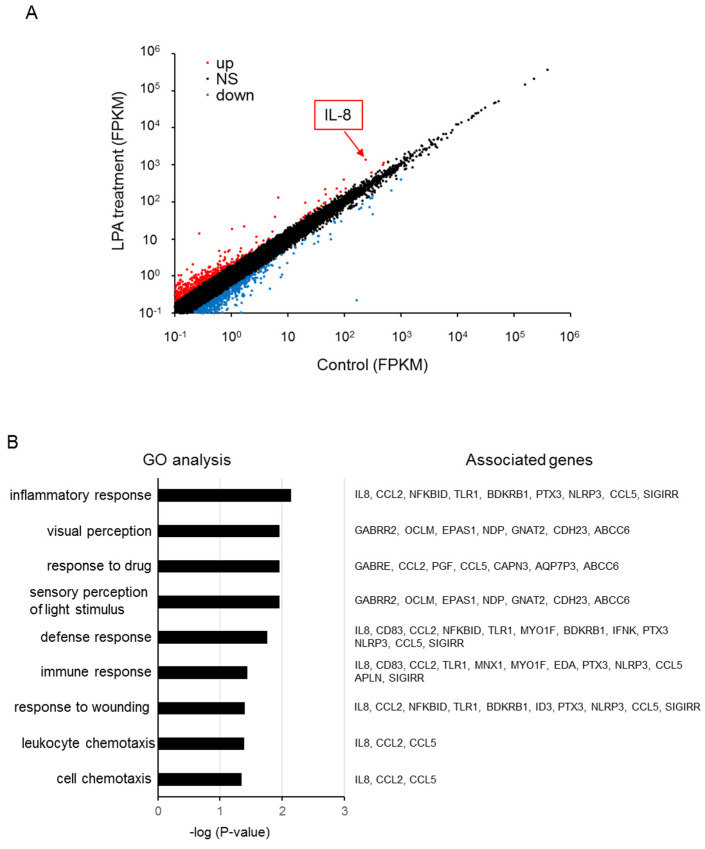
RNA-seq analysis of LPA-treated MDA-MB-231 cells. (**A**) Scatterplot showing the LPA-upregulated (red) and downregulated (blue) transcripts. (**B**) Gene ontology enrichment analysis of upregulated protein-coding genes.

**Figure 7 ijms-23-01967-f007:**
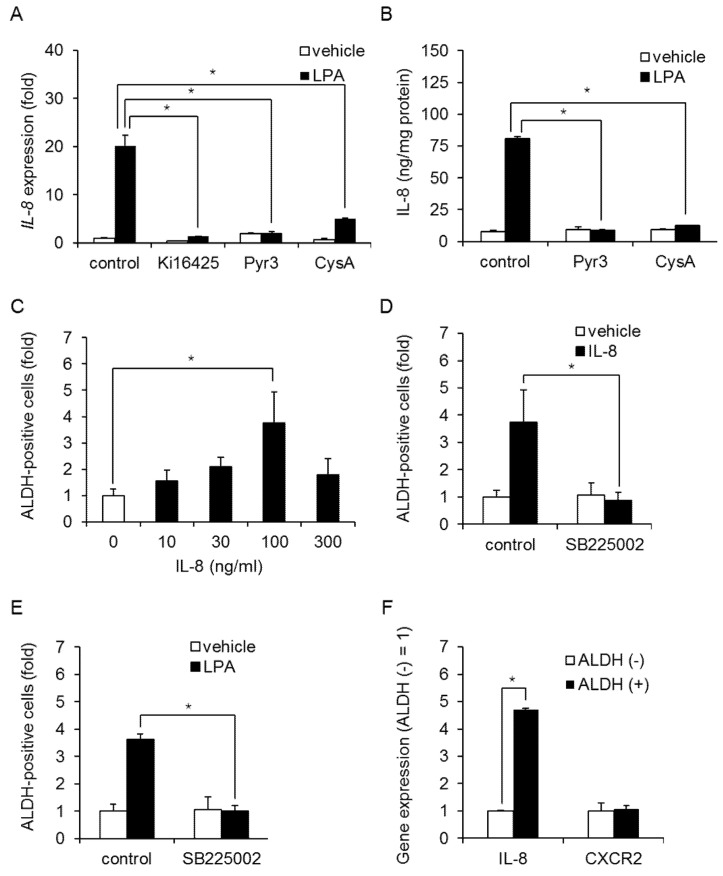
Role of IL-8 on LPA-induced increases in ALDH-positive cells in TNBC cells. (**A**) Effect of Ki16425 (10 µM), Pyr3 (1 µM), or CysA (10 µM) on LPA-induced IL-8 expression in MDA-MB-231 cells. Data represent the mean ± s.d. (*n* = 3). (**B**) Effect of Pyr3 (1 µM) or CysA (10 µM) on LPA-induced IL-8 secretion in MDA-MB-231 cells. The IL-8 content of the conditioned medium was measured by ELISA. Data represent the mean ± s.d. (*n* = 3). (**C**) Effect of IL-8 on the proportion of ALDH-positive cells in MDA-MB-231 cells. Data represent the mean ± s.d. (*n* = 3). (**D**) Effect of the IL-8 receptor antagonist SB225002 (1 µM) on the IL-8-induced proportion of ALDH-positive cells in MDA-MB-231 cells. Data represent the mean ± s.d. (*n* = 3). (**E**) Effect of the IL-8 receptor antagonist SB225002 (1 µM) on the LPA-induced proportion of ALDH-positive cells in MDA-MB-231 cells. Data represent the mean ± s.d. (*n* = 3). (**F**) IL-8 and CXCR2 expression in ALDH-positive and ALDH-negative cells by qPCR. Data represent the mean ± s.d. (*n* = 3). * *p* < 0.05.

**Figure 8 ijms-23-01967-f008:**
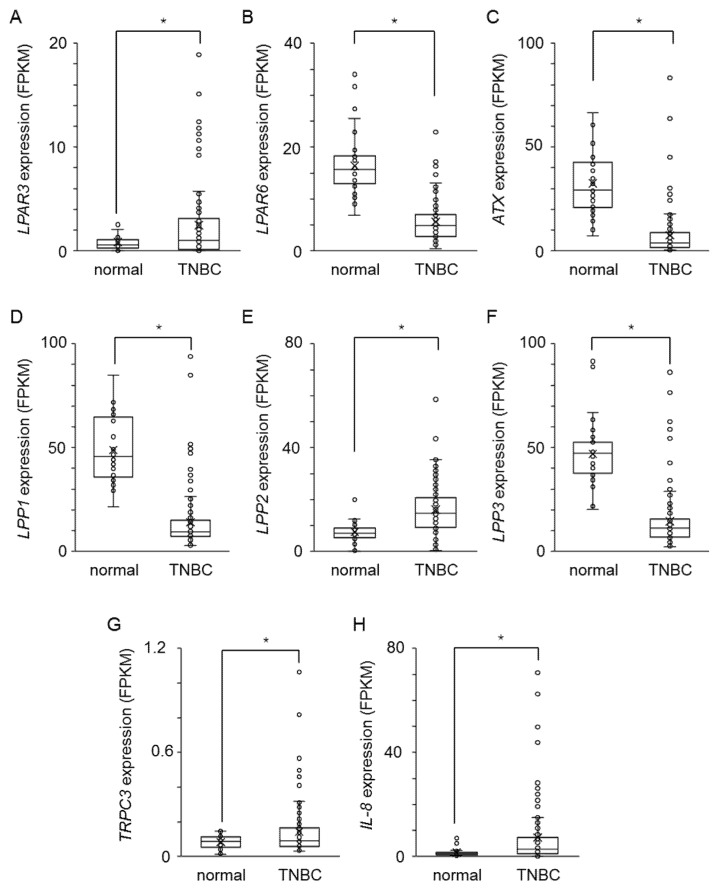
The clinical relevance of LPA signaling factors in human TNBC tissues. Expression level (FPKM) of LPAR3 (**A**), LPAR6 (**B**), ATX (**C**), LPP1 (**D**), LPP2 (**E**), LPP3 (**F**), TRPC3 (**G**), and IL-8 (**H**) in normal breast tissues (*n* = 38) and TNBC tissues (*n* = 115). * *p* < 0.05.

**Figure 9 ijms-23-01967-f009:**
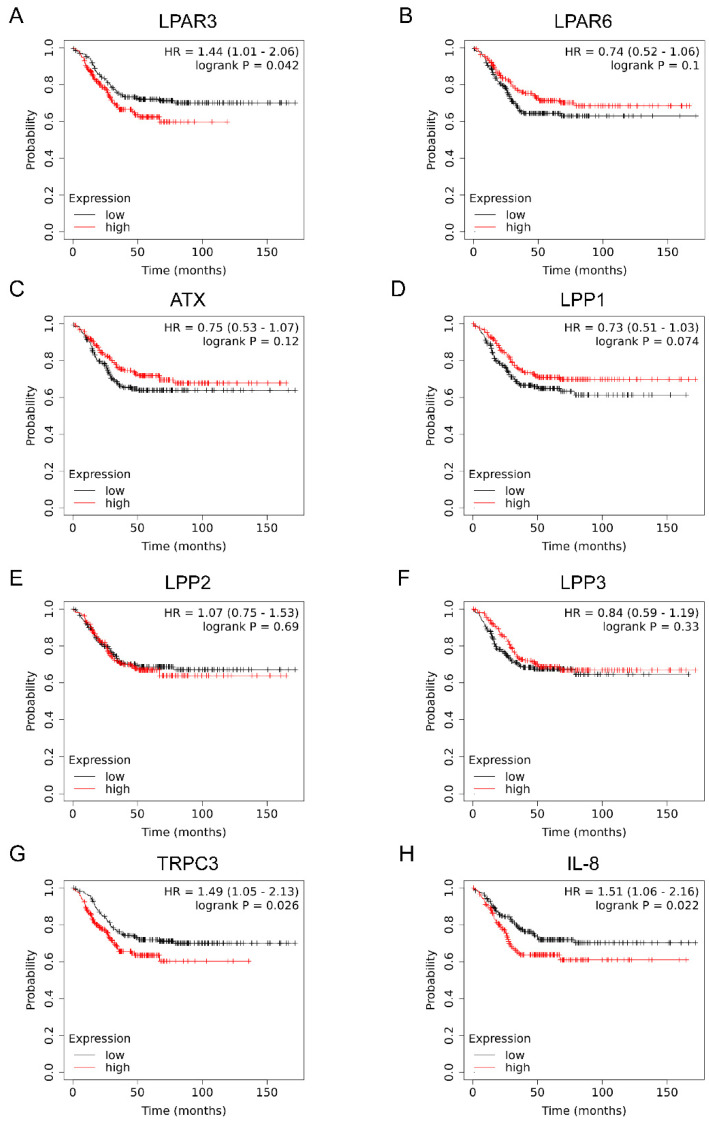
The effect of LPA signaling factors on the prognosis of TNBC patients. Distant metastasis free-survival of TNBC patients (*n* = 424) was analyzed using Kaplan–Meier Plotter (https://kmplot.com/analysis/index.php?p=background, accessed on 7 February 2022) according to expression of LPAR3 (**A**), LPAR6 (**B**), ATX (**C**), LPP1 (**D**), LPP2 (**E**), LPP3 (**F**), TRPC3 (**G**) and IL8 (**H**). The TNBC patient samples were divided into high expression and low expression groups by median. Hazard ratio (HR) and logrank P value are indicated in each panel.

**Figure 10 ijms-23-01967-f010:**
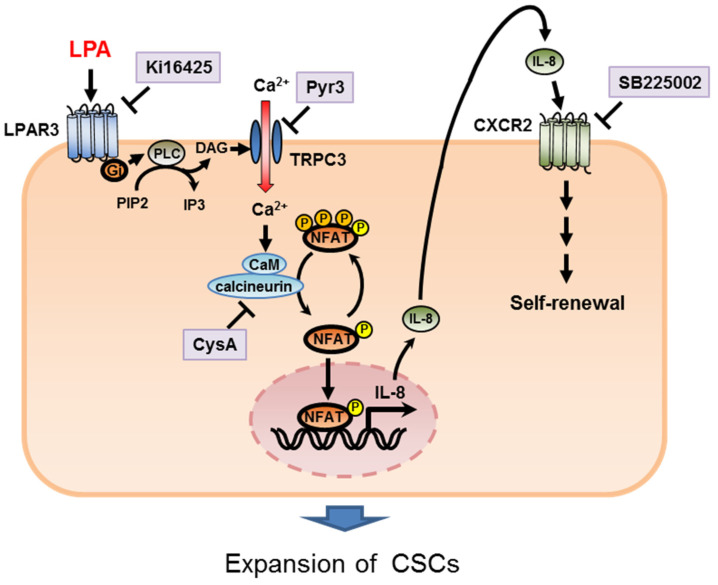
A working model of the functions of LPA/LPAR3/TRPC3 in CSC regulation.

**Table 1 ijms-23-01967-t001:** The table represents top ranked list of LPA-upregulated protein-coding genes.

Top 10 Upregulated Protein-Coding Genes in LPA Treated MDA-MB-231			
Gene_Short_Name	Control	LPA Treatment	ΔLPA (Fold Change)
GABRE	6.656	133.11	19.998
TNFAIP8L3	0.121	0.757	6.248
IL8	236.239	1403.039	5.939
KCTD4	0.113	0.666	5.901
HNRNPA1P33	0.635	3.529	5.554
IFITM4P	0.912	4.958	5.433
CAHM	0.105	0.549	5.207
LPL	0.102	0.488	4.790
HOXD4	20.308	96.461	4.750
BOC	0.233	1.083	4.652

## Data Availability

No data available.

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
