# Peer review of "Lysophosphatidic Acid Promotes the Expansion of Cancer Stem Cells via TRPC3 Channels in Triple-Negative Breast Cancer"

_ijms, 2022, doi:10.3390/ijms23041967_

Round 1

Reviewer 1 Report

This manuscript investigates the effect of lysophosphatidic acid (LPA) on cancer stem cell (CSC) expansion in triple negative breast cancer cells (TNBC) cells. The authors show that LPA enhances the number of CSCs and CSCs overexpress LPAR3 receptor. The LPAR1/3 antagonist (Ki16425) prevented CSC number and an LPAR3 agonist enhances CSC formation. Inhibition of LPAR2 or silencing of LPAR1 expression did not alter the increase in CSC number induced by LPA. LPA-induced calcium entry was prevented by treatment with Pyr3 and CSC formation was inhibited by Pyr3 as well as by PLC inhibition and TRPC3 silencing. LPA induces NFAT activation that is involved in IL-8 secretion, which, in turn, is involved in CSC formation. The issue investigated is interesting and the manuscript is carefully performed but some points need to be further confirmed to strengthen the authors hypothesis.

Major:

  1. In order to confirm the role of LPAR3 on LPA induced CSC formation, the authors should test the effect of silencing LPAR3 expression.
  2. The authors attribute the role of TRPC3 on LPA-induced calcium influx by using the pharmacological TRPC3 inhibitor Pyr3. To confirm the role of TRPC3 in calcium signaling the authors should test the effect of siTRPC3. It would also be important to assess the role of Orai1 in LPA-induced calcium entry as LPA induces NFAT activation and this event has been mostly associated to Orai1-mediated calcium entry.

Author Response

Responses to Reviewer 1 comment:

We would like to thank reviewers for careful reading our manuscript and several very constructive comments.

According to the reviewers' comments, we have revised the manuscript.

The revised points were written in red in the response letter and in the revised manuscript.

Our responses to the reviewers' reports are as follows.

Major

  1. In order to confirm the role of LPAR3 on LPA induced CSC formation, the authors should test the effect of silencing LPAR3 expression.

Answer:

According to the reviewer’s comment, we performed ALDH assay using LPAR3 siRNA. As shown in Figure 3C, LPAR3 siRNA inhibited the LPA-induced increase in ALDH-positive cells.  We confirmed that LPAR3 mediated LPA-induced increase in BCSC. Based on these data, we incorporated the following sentence into the manuscript:

(page 4, line118-120)

“Since Ki16425 inhibited both LPAR1 and LPAR3, we performed knockdown experiment. LPAR3 siRNA inhibited LPA-induced increase in ALDH-positive cells in MDA-MB-231 cells, whereas LPAR1 siRNA had little effect (Figure 3C).”

(page 5, line133-134)

“(C) Depletion of LPAR1 or LPAR3 with siRNA. Effect of LPAR1 or LPAR3 siRNA on the LPA-induced increase in ALDH-positive cells.”

  1. The authors attribute the role of TRPC3 on LPA-induced calcium influx by using the pharmacological TRPC3 inhibitor Pyr3. To confirm the role of TRPC3 in calcium signaling the authors should test the effect of siTRPC3.

Answer:

According to the reviewer’s comment, we examined the effect of siTRPC3 on LPA-induced calcium influx. As shown in Figure 4B, TRPC3 siRNA inhibited LPA-induced calcium influx. We confirmed that TRPC3 mediated LPA-induced calcium signaling. Based on these data, we incorporated the following sentence into the manuscript:

(page 6, line154-155)

“As shown in Figure 4B, treatment with Pyr3, a selective inhibitor of TRPC3 inhibited the LPA-induced increase in Ca2+ influx. TRPC3 siRNA produced similar results.”

(page 7, line169-170)

“(B) Effects of EGTA (1 mM), Pyr3 (1 µM) and TRPC3 siRNA on the LPA-induced increase in Ca2+ influx.”

It would also be important to assess the role of Orai1 in LPA-induced calcium entry as LPA induces NFAT activation and this event has been mostly associated to Orai1-mediated calcium entry.

Answer:

As you suggested, Pyr3 does not discriminate between Orai1 and TRPC3 channels (Schleifer H. et al. Br. J. Pharmacol. 2012). Orai1 has been shown to mediate LPA-induced Ca2+ influx and activation of NFAT2 (Jans R et al. J. Invest. Dermatol. 2013). We found that siTRPC3 completely inhibited LPA-induced both calcium entry and increase in ALDH-positive cells (Fig.4B and F). These data suggest that TRPC3 mediated LPA-induced proliferation of BCSCs in TNBC. Based on these data, we incorporated the following sentence into the manuscript:

(page 13, line289-page 14, line 292)

“Although Orai1 has been shown to mediate LPA-induced Ca2+ influx and activation of NFAT2 [31], we found that siTRPC3 completely inhibited LPA-induced both calcium entry and increase in ALDH-positive cells (Fig.4B and F), suggesting that TRPC3 mediated LPA-induced proliferation of BCSCs in TNBC.

(page 18, line 500-line 502)

“31. Jans, R.; Mottram, L.; Johnson, D.L.; Brown, A.M.; Sikkink, S.; Ross, K.; Reynolds, N.J. Lysophosphatidic Acid Promotes Cell Migration through STIM1- and Orai1-Mediated Ca2+i Mobilization and NFAT2 Activation. J. Invest. Dermatol. 2013, 133, 793–802.”

Reviewer 2 Report

Abstract: "the lipid mediator sphingosine-1-phosphate increased BCSCs in the estrogen receptor-positive cell line MCF-7, which were detected using aldehyde dehydrogenase (ALDH) activity-based flow-cytometry in TNBC cell line MDA-MB-231". This does not make sense. Needs re-worded.

Statistical analysis is missing for Fig. 1A. LPAR6 was also upregulated in ALDH-positive cells compared with ALDH-negative cells. The authors should explain why chose LPAR3.

Fig. 3C, LPAR3 and LPAR6 siRNA (or LPAR6 antagonist) are needed to explore the effect on CSC expansion.

Fig. 4A, Why the authors chose LPAR1/3 antagonist Ki16425, not selective LPAR3 agonist 2S-OMPT?

Fig. 4E-F, Primer sequence of TRPC3 for qPCR analysis is missed.

Fig. 6A, a Heatmap can better show differentially expressed genes after LPA treatment.
All the RNA-seq analysis data should be given, or deposited in the public database (such GEO).

Fig. 7A,F, Primer sequences of IL8 and CXCR2 for qPCR analysis are missed.

Fig. 8, Detail information about RNA-seq data of breast cancer patient obtained from GDC data portal should be provided. Since I noticed there are only 38 normal breast tissues and 115 TNBC tissues. 
The effect of those genes on the prognosis of TNBC patients should also be investigated using public clinical data.

Author Response

Responses to Reviewer 2 comment:

We would like to thank reviewers for careful reading our manuscript and several very constructive comments.

According to the reviewers' comments, we have revised the manuscript.

The revised points were written in red in the response letter and in the revised manuscript.

Our responses to the reviewers' reports are as follows.

Comments and Suggestions for Authors

  1. Abstract: "the lipid mediator sphingosine-1-phosphate increased BCSCs in the estrogen receptor-positive cell line MCF-7, which were detected using aldehyde dehydrogenase (ALDH) activity-based flow-cytometry in TNBC cell line MDA-MB-231 ". This does not make sense. Needs re-worded.

Answer:

We are sorry for the confusion. According to the reviewer’s suggestion, we modified the sentence as follows:

(page 1, line20-2

“We have previously reported that the lysosphingolipid sphingosine-1-phosphate mediates the CSC phenotype, which can be identified as the ALDH-positive cell population in several types of human cancer cell lines.”

  1. Statistical analysis is missing for Fig. 1A. LPAR6 was also upregulated in ALDH-positive cells compared with ALDH-negative cells. The authors should explain why chose LPAR3.

Answer:

As the reviewer suggested, LPAR6 was also upregulated in ALDH-positive cells, compared with ALDH-negative cells (Figure 1A). We added the graph of LPAR3 and LPAR6 expression (Figure S1). Expression level of LPAR3 in ALDH-positive cells was 13.2-fold higher than that of LPAR6 in ALDH-positive cells. Therefore, we focused on LPAR3 in BCSCs. We added Supplementary Figure 1 and incorporated the following sentence into the manuscript:

(page 2, line84-87)

“LPAR3 and LPAR6 expression was upregulated in ALDH-positive cells compared with ALDH-negative cells (Figure 1A). Since expression level of LPAR3 in ALDH-positive cells was 13.2-fold higher than that of LPAR6 in ALDH-positive cells (Figure S1), we focused on LPAR3 in BCSCs.”

  1. Fig. 3C, LPAR3 and LPAR6 siRNA (or LPAR6 antagonist) are needed to explore the effect on CSC expansion.

Answer:

According to the reviewer’s comment, we performed ALDH assay using LPAR3 specific siRNA. Since LPAR6 expression was very lower than LPAR3, we performed only LPAR3 knockdown. As shown Figure 3C, LPAR3 siRNA completely inhibited LPA-induced increase in ALDH-positive cells. We confirmed that LPAR3 siRNA did not affect LPAR6 expression (Figure. S2B). LPA-induced increase in BCSCs is considered to be mediated by LPAR3, not by LPAR6. We incorporated these results into Figure 3C and Figure S2B. We incorporated the following sentence into the manuscript:

(page 4, line118-122)

“Since Ki16425 inhibited both LPAR1 and LPAR3, we performed knockdown experiment. LPAR3 siRNA inhibited LPA-induced increase in ALDH-positive cells in MDA-MB-231 cells, whereas LPAR1 siRNA had little effect (Figure 3C). LPAR3 siRNA did not affect LPAR6 expression (Figure S2B). These data suggest that LPAR3 mediates LPA-induced increase in BCSCs.”

(page 5, line133-134)

“(C) Depletion of LPAR1 or LPAR3 with siRNA. Effect of LPAR1 or LPAR3 siRNA on the LPA-induced increase in ALDH-positive cells.”

  1. Fig. 4A, Why the authors chose LPAR1/3 antagonist Ki16425, not selective LPAR3 agonist 2S-OMPT?

Answer:

According to the reviewer’s comment, we added the effect of LPAR3 selective agonist 2S-OMPT on calcium influx. As shown in Figure 4A, 2S-OMPT increased calcium influx. We incorporated the following sentence into the manuscript:

(page 6, line148-150)

“We found that LPA or 2SOMPT stimulation increased intracellular Ca2+, and the LPAR1/3 antagonist Ki16425 inhibited the LPA-induced increase in Ca2+ influx in MDA-MB-231 cells (Figure 4A).”

(page 7, line166-167)

“(A) After stimulation with LPA (10 µM) or 2S-OMPT (3 µM), intracellular Ca2+ in bulk cells was measured using Fluo-4.”

  1. Fig. 4E-F, Primer sequence of TRPC3 for qPCR analysis is missed.

Answer:

According to the reviewer’s comment, we added primer sequence of TRPC3 in Supplementary Table 2.

  1. Fig. 6A, a Heatmap can better show differentially expressed genes after LPA treatment.

All the RNA-seq analysis data should be given, or deposited in the public database (such GEO).

Answer:

According to the reviewer’s comment, we added all the RNA-seq analysis data (FPKM value of control and LPA treatment) in Supplementary Table 1. We incorporated the following sentence into the manuscript:

(page 8, line197-199)

“As shown in Figure 6A and Table S1, we found that 428 transcripts were upregulated > 2-fold and 420 transcripts were downregulated < 0.5-fold among 15696 transcripts (> 0.1 FPKM).”

(page 15, line419-420)

“Table S1: RNA-seq analysis of LPA-treated MDA-MB-231 cells,”

  1. Fig. 7A,F, Primer sequences of IL8 and CXCR2 for qPCR analysis are missed.

Answer:

According to the reviewer’s comment, we added primer sequence of IL-8 and CXCR2 in Supplementary Table 2.

Fig. 8, Detail information about RNA-seq data of breast cancer patient obtained from GDC data portal should be provided. Since I noticed there are only 38 normal breast tissues and 115 TNBC tissues. The effect of those genes on the prognosis of TNBC patients should also be investigated using public clinical data.

Answer:

We apologize for the insufficient explanation about RNA-seq data of breast cancer patient obtained from GDC data portal. We selected 115 TNBC patient data from TCGA-BRCA dataset. Based on “ER status by IHC”, “PR status by IHC” and “HER2 status by IHC”, we selected all negative status (ER, PR, HER2 negative) as TNBC patients. In addition, we obtained 38 available normal breast tissues tissue data. We added the following sentence into the manuscript:

(page 16, line 390-394)

“RNA-seq data of breast cancer patients were downloaded from the National Cancer Institute GDC data portal. We selected 115 TNBC patient data from TCGA-BRCA dataset. Based on “ER status by IHC”, “PR status by IHC” and “HER2 status by IHC”, we selected all negative status (ER, PR, HER2 negative) as TNBC patients. In addition, we obtained 38 available normal breast tissues tissue data.”

The effect of those genes on the prognosis of TNBC patients should also be investigated using public clinical data.

Answer:

According to reviewer’s suggestion, we analyzed the effect of LPA signaling factors on the prognosis of TNBC patients. As shown in Figure 9, mRNA level of LPAR3, TRPC3 and IL-8 were reversely correlated with the distant metastasis free-survival (DMFS) of TNBC patients. We added Figure 9 and incorporated the following sentence into the manuscript:

(page 11, line 247-page 12, line 250)

“We then analyzed the effect of these genes on the prognosis of TNBC patients using Kaplan-Meier plotter. As shown in Figure 9, mRNA level of LPAR3, TRPC3 and IL-8 were reversely correlated with the distant metastasis free-survival of TNBC patients. In contrast, LPAR6, ATX, LPP1-3 had little effect on distant metastasis free-survival.”

(page 12, line 256-259)

Figure 9. The effect of LPA signaling factors on the prognosis of TNBC patients. Distant metastasis free-survival of TNBC patients (n=424) was analyzed using Kaplan-Meier Plotter. The TNBC patient samples were divided into high expression and low expression groups by median. Hazard ratio (HR) and logrank P value are indicated in each panel.”

(page 14, line 302-303)

“Clinical data analysis supports prognostic role of TRPC3 and IL-8 in TNBC patients (Figure 9).”

(page 16, line 394-398)

The effect of LPA signaling factors on DMFS of TNBC patients (ER-negative, PR-negative, and HER2-negative, n=424) was analyzed using Kaplan-Meier Plotter (https://kmplot.com/analysis/index.php?p=background). The TNBC patient samples were divided into high expression and low expression groups by median.

Round 2

Reviewer 1 Report

The authors have adequately addressed my previous concerns and improved the manuscript.

Reviewer 2 Report

The authors have addressed majority of my comments, and I'd recommend the manuscript for publication at this time.